# Learning what you can do
# before doing anything

**Oleh Rybkin**[*,1]**, Karl Pertsch**[*,2]
**Konstantinos G. Derpanis**[3,4]**, Kostas Daniilidis**[1]**, Andrew Jaegle**[1]
[1]University of Pennsylvania
[2]University of Southern California
[3]Ryerson University
[4]Samsung AI Centre Toronto

## Abstract

Intelligent agents can learn to represent the action spaces of other agents simply by observing them act. Such representations help agents quickly learn to predict the effects of their own actions on the environment and to plan complex action sequences. In this work, we address the problem of learning an agent's action space purely from visual observation. We use stochastic video prediction to learn a latent variable that captures the scene's dynamics while being minimally sensitive to the scene's static content. We introduce a loss term that encourages the network to capture the composability of visual sequences and show that it leads to representations that disentangle the structure of actions. We call the full model with composable action representations Composable Learned Action Space Predictor (CLASP). We show the applicability of our method to synthetic settings and its potential to capture action spaces in complex, realistic visual settings. When used in a semi-supervised setting, our learned representations perform comparably to existing fully supervised methods on tasks such as action-conditioned video prediction and planning in the learned action space, while requiring orders of magnitude fewer action labels.[1]

## 1 Introduction

Agents behaving in real-world environments rely on perception to judge what actions they can take and what effect these actions will have. Purely perceptual learning may play an important role in how these action representations are acquired and used. In this work, we focus on the problem of learning an agent's action space from unlabeled visual observations. To see the usefulness of this strategy, consider an infant that is first learning to walk. From around 10 months of age, infants rapidly progress from crawling, to irregular gaits with frequent falling, and finally to reliable locomotion (Adolph et al. (2012)). But before they first attempt to walk, infants have extensive sensory exposure to adults walking. Unsupervised learning from sensory experience of this type appears to play a critical role in how humans acquire representations of actions before they can reliably reproduce the corresponding behaviour (Ullman et al. (2012)). Infants need to relate the set of motor primitives they can generate to the action spaces exploited by adults (Dominici et al. (2011)), and a representation acquired by observation may allow an infant to more efficiently learn to produce natural, goal-directed walking behavior.

Reinforcement learning (RL) provides an alternative to the (passive) unsupervised learning approach as it implicitly discovers an agent's action space and the consequences of its actions. Recent breakthroughs in model-free and model-based RL suggest that end-to-end training can be used to learn mappings between sensory input and actions (Mnih et al. (2015); Lillicrap et al. (2016); Levine et al. (2016); Finn & Levine (2017); Schulman et al. (2015)). However, these methods require active observations and the sensorimotor mappings learned in this way cannot be easily generalized to new agents with different control interfaces. Methods for sensorimotor learning from purely visual

---

[*]Equal contribution. Ordering determined by a coin flip.
[1]Project website: `https://daniilidis-group.github.io/learned_action_spaces`

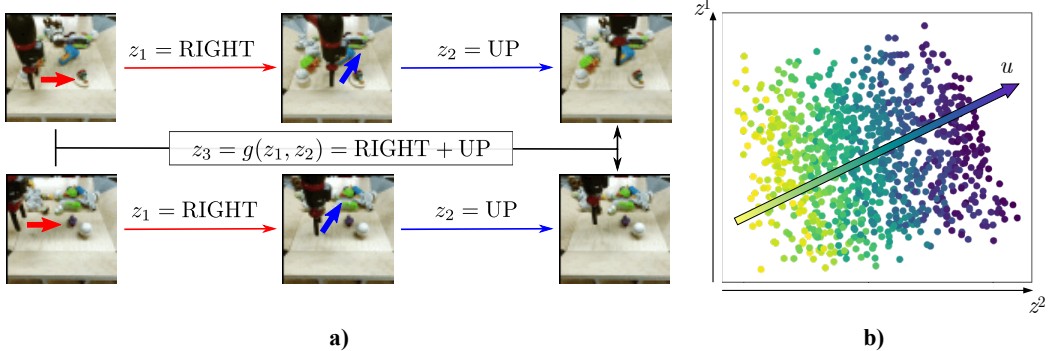

Figure 1: Using latent composition to recover actions from passive data. **a)** Two sequences starting from different initial states but changing according to the same actions. Without requiring labels, our model learns to represent the action in sequences like these identically. We train a representation $z$ to capture the dynamics of the scene and its compositional structure: applying ($z_1$ and $z_2$) should have the same effect as applying the composed representation $g(z_1, z_2)$. These properties capture the fact that effector systems, such as a robot arm, use the same composable action space in many different states. **b)** The learned action space $z$ recovered by our method (PCA visualization). Points are colored by the true action $u$: true actions can be easily decoded from $z$, validating that the structure of the action space has been captured.

data may facilitate learning where action information is not available, such as when using video data collected from the Internet. Such methods may also be useful for imitation learning, where ground truth actions are often hard or impossible to collect other than by visual observation (Finn et al. (2017); Pathak et al. (2018)). More generally, learning from passive observations may make it easier to reuse action representations between systems with different effectors and goals. The representations learned by unsupervised methods are invariant to these choices because the model does not have access to motor commands or goals during training.

In this work, we evaluate the proposal that learning *what you can do before doing anything* can lead to action space representations that make subsequent learning more efficient. To this end, we develop a model that learns to represent an agent's action space given only unlabeled videos of the agent. The resulting representation enables direct planning in the latent space. Given a small number of action-labeled sequences we can execute the plan by learning a simple mapping from latent action representations to the agent's controls. This representation may be analogous to those in the parietal and premotor areas of cortex, which include populations of neurons that represent the structure of actions produced both by the self and by others (Rizzolatti et al. (1996); Romo et al. (2004)) and that are critical for reliably producing flexible, voluntary motor control (see Kandel et al. (2012), Chapter 38). In the brain, representations of this kind could plausibly be learned using specialized loss functions (Marblestone et al. (2016)) whose effect is to induce the prior needed to determine the structure of actions in observation data.

In contrast to most approaches to unsupervised learning of dynamics, which focus on learning the statistical structure of the environment, we focus on disentangling action information from the instantaneous state of the environment (Fig. 1). We base our work on recent stochastic video prediction methods (Babaeizadeh et al. (2018); Denton & Fergus (2018); Lee et al. (2018)) and impose two properties on the latent representation. First, we train the representation to be *minimal*, i.e. containing minimal information about the current world state. This forces the representation to focus on dynamic properties of the sensory input. A similar objective has been used in previous work to constrain the capacity of video prediction models (Denton & Fergus (2018)). Second, we train the representation to be *composable* by introducing a novel loss term that enforces that the cumulative effect of a sequence of actions can be computed from the individual actions' representations (Fig. 1, left). Composability encourages disentangling: as a composed representation does not have access to the static content of the intermediate frames, a representation is composable only if the individual action representations are disentangled from the static content. Taken together, these two properties lead to a representation of sensory dynamics that captures the structure of the agent's actions.

We make the following three contributions. First, we introduce a method for unsupervised learning of an agent's action space by training the latent representation of a stochastic video prediction model for the desiderata of minimality and composability. Second, we show that our method learns a representation of actions that is independent of scene content and visual characteristics on (i) a simulated robot with one degree of freedom and (ii) the BAIR robot pushing dataset (Ebert et al. (2017)). Finally, we demonstrate that the learned representation can be used for action-conditioned video prediction and planning in the learned action space, while requiring orders of magnitude fewer action-labeled videos than extant supervised methods.

## 2 RELATED WORK

**Learning structured and minimal representations.**   Several groups have recently shown how an adaptation of the variational autoencoder (VAE, Kingma & Welling (2014); Rezende et al. (2014)) can be used to learn representations that are minimal in the information-theoretic sense. Alemi et al. (2017) showed that the Information Bottleneck (IB) objective function (Tishby et al. (1999); Shwartz-Ziv & Tishby (2017)) can be optimized with a variational approximation that takes the form of the VAE objective with an additional weighting hyperparameter. In parallel, Higgins et al. (2017) showed that a similar formulation can be used to produce disentangled representations. The connection between disentaglement and minimality of representations was further clarified by Burgess et al. (2018). In this work, we apply the IB principle to temporal models to enforce minimality of the representation.

Several groups have proposed methods to learn disentangled representations of static content and pose from video (Denton & Birodkar (2017); Tulyakov et al. (2018)). Jaegle et al. (2018) learn a motion representation by enforcing that the motion acts on video frames as a group-theoretic action. In contrast, we seek a representation that disentangles the motion from the static pose.

Thomas et al. (2017) attempt to learn a disentangled representation of controllable factors of variation. While the goals of their work are similar to ours, their model relies on active learning and requires an embodied agent with access to the environment. In contrast, our model learns factors of variation purely from passive temporal visual observations, and thus can be applied even if access to the environment is costly or impossible.

**Unsupervised learning with video data.**   Several recent works have exploited temporal information for representation learning. Srivastava et al. (2015) used the Long Short-Term Memory (LSTM, Hochreiter & Schmidhuber (1997)) recurrent neural network architecture to predict future frames and showed that the learned representation was useful for action recognition. Vondrick et al. (2016) showed that architectures using convolutional neural networks (CNNs) can be used to predict actions and objects several seconds into the future. Recently, work such as Finn et al. (2016); Villegas et al. (2017); Denton & Birodkar (2017) has proposed various modifications to the convolutional LSTM architecture (Xingjian et al. (2015)) for the task of video prediction and shown that the resulting representations are useful for a variety of tasks.

Others have explored applications of video prediction models to RL and control (Weber et al. (2017); Ha & Schmidhuber (2018); Wayne et al. (2018)). Chiappa et al. (2017) and Oh et al. (2015) propose models that predict the consequences of actions taken by an agent given its control output. Similar models have been used to control a robotic arm (Agrawal et al. (2016); Finn & Levine (2017); Ebert et al. (2017)). The focus of this work is on learning action-conditioned predictive models. In contrast, our focus is on the unsupervised discovery of the space of possible actions from video data.

Our model is inspired by methods for stochastic video prediction that, given a sequence of past frames, capture the multimodal distribution of future images (Goroshin et al. (2015); Henaff et al. (2017)). We use the recently proposed recurrent latent variable models based on the variational autoencoder (Babaeizadeh et al. (2018); Denton & Fergus (2018); Lee et al. (2018)). We develop these methods and propose a novel approach to unsupervised representation learning designed to capture an agent's action space.

**Sensorimotor abstractions for behavior.**   There is a long history of work developing sensorimotor representations for applications in reinforcement learning and robotics. Previous work in this domain has primarily focused on introducing hand-crafted abstractions and hierarchies to make sensorimotor mappings more generalizable. Methods for aggregating low-level controls into higher-

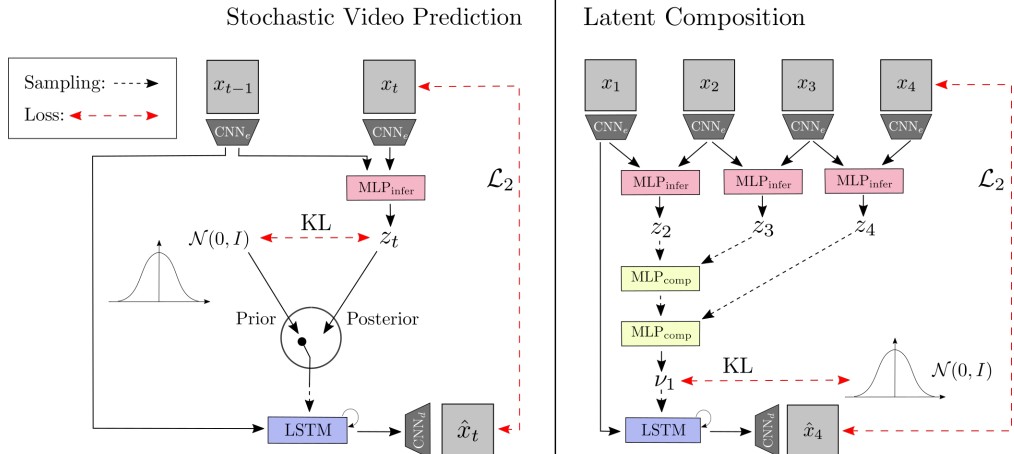

Figure 2: Components of the proposed architecture. **Left**: The stochastic video prediction model, shown for one timestep. During training, we estimate the latent variable $z_t$ using the approximate inference network (MLP$_\text{infer}$, CNN$_e$) from the current and previous image. At test time, we produce $z_t$ using the prior distribution $p(z) \sim \mathcal{N}(0, I)$. Future frames are estimated by passing $z_t$ together with images $x_{t-1}$ through the generative network (LSTM, CNN$_d$). Please refer to Appendices A and B for architectural details. **Right**: Composability training. Latent samples $z$ are concatenated pairwise and passed through the composition network MLP$_\text{comp}$ that defines a distribution over $\nu$ in the trajectory space. A sampled value of $\nu$ is decoded into an image through the same generative network (LSTM and CNN$_d$) and matched to the final image in the composed sequence.

level representations on which planning and RL can be performed are well-studied in the robotics literature: notable examples include the options framework for hierarchical RL (Sutton et al. (1999); Bacon et al. (2017)), dynamic motion primitives for manipulation (Schaal et al. (2005); Schaal (2006); Niekum et al. (2015)), and recent work that abstracts a learned policy away from low-level control and perception to ease simulation-to-real transfer (Clavera & Abbeel (2017); Müller et al. (2018)). Other work has learned to separate robot-instance specific controls from task-related skills through modular policies, but this work does not enforce any structure onto the intermediate representation and requires extensive interaction with the environment (Devin et al. (2017)).

## 3 APPROACH

In this section, we describe our architecture for learning an action representation that is minimal and composable. In Sec. 3.1, we describe a variational video prediction model similar to that of Denton & Fergus (2018) that provides us with a framework for learning a latent representation $z_t$ at time $t$ of the change between the past and the current frames. No labeled actions are considered at this stage. In Sec. 3.2, we introduce an unsupervised method for imposing composability of the latent that allows us to recover a structured representation that defines CLASP. To verify that the learned representation corresponds to the executed control, we show that we can learn a bijective mapping between the latent representation and the control output executed at that time using a small number of labeled data points (Sec. 3.3). In the experimental section, we describe how the learned bijective mapping can be used for tasks such as action-conditioned video prediction (Sec. 4.2) and planning in the learned action space (Sec. 4.3).

### 3.1 VIDEO PREDICTION MODEL

At the core of our method is a recurrent latent variable model for video prediction based on a temporal extension of the conditional VAE proposed by Chung et al. (2015). We consider the generative model shown in Fig. 2 (left). At each timestep $t$, the model outputs a latent variable $z_t \sim p(z) = \mathcal{N}(0, I)$ associated with this timestep. Given a history of frames $\mathbf{x}_{1:t-1}$ and latent samples $\mathbf{z}_{2:t}$, the generative distribution over possible next frames is given by $x_t \sim p_\theta(x_t | \mathbf{x}_{1:t-1}, \mathbf{z}_{2:t}) = \mathcal{N}(\mu_\theta(\mathbf{x}_{1:t-1}, \mathbf{z}_{2:t}), I)$.

In practice, we generate the next frame by taking the mean of the conditional distribution: $\hat{x}_t = \mu_\theta(\mathbf{x}_{1:t-1}, \mathbf{z}_{2:t})$.

To optimize the log-likelihood of this generative model, we introduce an additional network approximating the posterior of the latent variable $z_t \sim q_\phi(z_t|x_t, x_{t-1}) = \mathcal{N}(\mu_\phi(x_t, x_{t-1}), \sigma_\phi(x_t, x_{t-1}))$. We can optimize the model using the variational lower bound of the log-likelihood in a formulation similar to the original VAE. However, as has been shown recently by Alemi et al. (2018), the standard VAE formulation does not constrain the amount of information contained in the latent variable $z$. To overcome this, and to learn a minimal representation of $z$, we reformulate the standard VAE objective in terms of the Information Bottleneck (IB) (Shwartz-Ziv & Tishby (2017)).

IB minimizes the mutual information, $I$, between the action representation, $z_t$, and input frames, $\mathbf{x}_{t-1:t}$, while maximizing the ability to reconstruct the frame $x_t$ as measured by the mutual information between $(z_t, x_{t-1})$ and $x_t$:

$$\max_{p_\theta, q_\phi} I((z_t, x_{t-1}), x_t) - \beta_z I(z_t, x_{t-1:t}). \tag{1}$$

The two components of the objective are balanced with a Lagrange multiplier $\beta_z$. When the value of $\beta_z$ is higher, the model learns representations that are more efficient, i.e. minimal in the information-theoretic sense. We use this property to achieve our first objective of *minimality* of $z$.

The variational IB (Alemi et al. (2017)) provides a variational approximation of the IB objective, that simply takes the form of the original VAE objective with an additional constant $\beta_z$. Aggregating over a sequence of frames, the video prediction objective for our model is given by:

$$\mathcal{L}_{\theta,\phi}^{pred}(\mathbf{x}_{1:T}) = \sum_{t=1}^{T} \left[ \mathbb{E}_{q_\phi(\mathbf{z}_{2:t}|\mathbf{x}_{1:t})} \log p_\theta(x_t|\mathbf{x}_{1:t-1}, \mathbf{z}_{2:t}) - \beta_z D_{KL}(q_\phi(z_t|\mathbf{x}_{t-1:t})||p(z)) \right]. \tag{2}$$

The full derivation of the variational lower bound is given in the appendix of Denton & Fergus (2018)[2]. The full model for one prediction step is shown in the left part of Fig. 2.

### 3.2 CLASP: LEARNING ACTION REPRESENTATIONS WITH COMPOSABILITY

Given a history of frames, the latent variable $z_t$ represents the distribution over possible next frames. It can thus be viewed as a representation of possible changes between the previous and the current frame. We will associate the latent variable $z_t$ with the distribution of such changes. In video data of an agent executing actions in an environment, the main source of change is the agent itself. Our model is inspired by the observation that a natural way to represent $z_t$ in such settings is by the agents' actions at time $t$. In this section, we describe an objective that encourages the previously described model (Sec. 3.1) to learn action representations.

To encourage *composability* of action representations, we use the procedure illustrated in Fig. 2 (right). We define an additional random variable $\nu_t \sim q_\zeta(\nu_t|z_t, z_{t-1}) = \mathcal{N}(\mu_\zeta(z_t, z_{t-1}), \sigma_\zeta(z_t, z_{t-1}))$ that is a representation of the trajectory $z_{t-1:t}$. The process of composing latent samples into a single trajectory can be repeated several times in an iterative fashion, where the inference model $q_\zeta$ observes a trajectory representation $\nu_{t-1}$ and the next latent $z_t$ to produce the composed trajectory representation $\nu_t \sim q_\zeta(\nu_t|\nu_{t-1}, z_t)$. The inference model $q_\zeta$ is parameterized with a multilayer perceptron, MLP$_{\text{comp}}$.

We want $\nu$ to encode entire trajectories, but we also require it to have minimal information about individual latent samples. We can encourage these two properties by again using the IB objective:

$$\max_{p_\theta, q_\phi, \zeta} I((\nu_t, x_1), x_t) - \beta_\nu I(\mathbf{z}_{2:t}, \nu_t). \tag{3}$$

We maximize this objective using the following procedure. Given a trajectory of $T$ frames, we use MLP$_{\text{infer}}$ to retrieve the action representations $z$. Next, we generate a sequence of trajectory representations $\nu_t$, each of which is composed from $C$ consecutive action representations $\mathbf{z}_{t-C:t}$. We obtain $T_C = \lfloor T/C \rfloor$ such representations. Finally, we use $\nu_t$ to produce the corresponding frames

---

[2]Denton & Fergus (2018) use the objective with $\beta_z$, but formulate this objective in terms of the original VAE.

$\hat{x}_t = p_\theta(x_t|x_{t-C}, \nu_t)^3$. The variational approximation to (3) that we use to impose composability takes the following form:

$$\mathcal{L}_{\theta,\phi,\zeta}^{comp}(\mathbf{x}_{1:T}) = \sum_{t=1}^{T_C} \left[ \mathbb{E}_{q_{\phi,\zeta}(\nu_{1:t}|\mathbf{x}_{1:T})} \log p_\theta(x_{t \times T_c}|\mathbf{x}_{1:(t-1) \times T_C}, \nu_{1:t}) \right. \tag{4}$$
$$\left. - \beta_\nu D_{KL}(q_{\phi,\zeta}(\nu_t|\mathbf{x}_{(t-1) \times T_C : t \times T_C})||p(\nu)) \right],$$

where the prior distribution over $\nu$ is given by the unit Gaussian $\nu \sim p(\nu) = \mathcal{N}(0, I)$.

The objective above encourages the model to find a minimal representation for the trajectories $\nu$. As the trajectories are composed from only the action representations $z$, this encourages $z$ to assume a form suitable for efficient composition. This allows us to recover an action representation that is *composable*. Our overall training objective is the sum of the two objectives:

$$\mathcal{L}_{\theta,\phi,\zeta}^{total} = \mathcal{L}_{\theta,\phi,\zeta}^{comp} + \mathcal{L}_{\theta,\phi}^{pred}. \tag{5}$$

We call the full model with composable action representations Composable Learned Action Space Predictor (CLASP).

### 3.3 Grounding the control mapping

Our approach allows us to learn a latent representation $z$ that is minimal and disentangled from the content of previous images. To use such a learned representation for control, we want to know which action $u$ a certain sample $z$ corresponds to, or vice versa. To determine this correspondence, we learn a simple bijective mapping from a small number of action-annotated frame sequences from the training data. We train the bijection using two lightweight multilayer perceptrons, $\hat{z}_t = \text{MLP}_{\text{lat}}(u_t)$ and $\hat{u}_t = \text{MLP}_{\text{act}}(z_t)$. Note that only the $\text{MLP}_{\text{lat}}$ and $\text{MLP}_{\text{act}}$ networks are trained in this step, as we *do not* propagate the gradients into the video prediction model. Because we do not have to re-train the video prediction model, this step requires far less data than models with full action supervision (Section 4.3).

We note that standard image-based representations of motion, e.g., optical flow, do not directly form a bijection with actions in most settings. For example, the flow field produced by a reacher (as in Fig. 5) rotating from 12 o'clock to 9 o'clock is markedly different from the flow produced by rotating from 3 o'clock to 12 o'clock, even though the actions producing the two flow fields are identical (a 90 degree counter-clockwise rotation in both cases). In contrast, our representation easily learns a bijection with the true action space.

## 4 Empirical evaluation

For evaluation, we consider tasks that involve regression from the latent variable $z$ to actions $u$ and vice versa. By learning this bijection we show that our model finds a representation that directly corresponds to actions and is disentangled from the static scene content. We show that after CLASP is trained, it can be used for both action-conditioned video prediction and planning (see Fig. 4), and provide a procedure to plan in the learned representation. We also validate that our approach requires orders of magnitude fewer labels than supervised approaches, and that it is robust to certain visual characteristics of the agent or the environment. Please refer to Appendix B for the exact architectural parameters.

**Datasets.** We conduct experiments on a simple simulated reacher dataset and the real-world Berkeley AI Research (BAIR) robot pushing dataset from Ebert et al. (2017). The reacher dataset consists of sequences of a robot reacher arm with one degree of freedom rotating counter-clockwise with random angular distances between consecutive images. We simulate it using OpenAI's Roboschool environment (Klimov & Schulman (2018)). The actions $u$ are encoded as relative angles between images, and constrained to the range $u \in [0°, 40°]$. The dataset consists of $100\,000$ training and $4000$

---

[3]To allow the generative model to distinguish between individual action representations $z$ and trajectory representations $\nu$, we concatenate them with a binary indicator set to 0 for $z$ and 1 for $\nu$. With the binary indicator, we can control whether the generative network interprets an input latent as the representation of a single action or a whole trajectory.

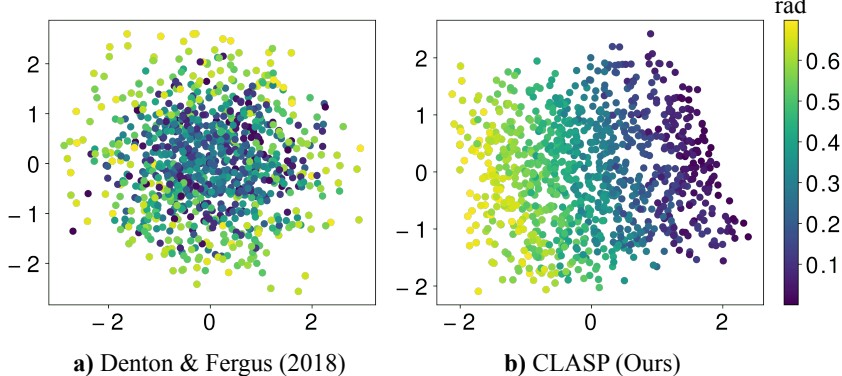

**a)** Denton & Fergus (2018)          **b)** CLASP (Ours)

Figure 3: Visualization of the learned action space, $z$, on the reacher dataset. Each of the 1000 points depicts a value of $z$ for a different frame pair from the dataset. We plot the projection of $z$ onto the first two principal components of the data. Each point is colored by the value of the ground truth rotation, in radians, depicted in the two images used to infer $z$ for that point. **a)** The latent space learned by the baseline model has no discernible correspondence to the ground truth actions. **b)** Our method learns a latent space with a clear correspondence to the ground truth actions. In the Appendix, Fig. 15 further investigates why the baseline fails to produce a disentangled representation.

test sequences. Additionally, we create two variations of this dataset, with (i) varying backgrounds taken from the CIFAR-10 dataset (Krizhevsky (2009)) and (ii) varying robot appearance, with 72 different combinations of arm length and width in the training dataset.

The BAIR robot pushing dataset comprises $44\,374$ training and $256$ test sequences of 30 frames each from which we randomly crop out subsequences of 15 frames. We define actions, $u$, as differences in the spatial position of the end effector in the horizontal plane[4].

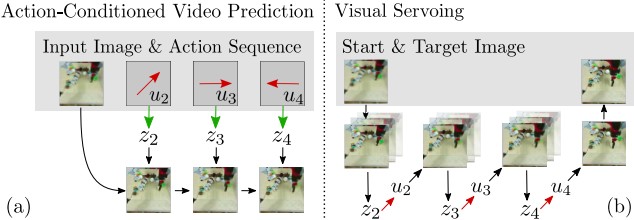

(a)          (b)

**Baselines.** We compare to the original model of Denton & Fergus (2018) that does not use the proposed composability objective. To obtain an upper bound on our method's performance we also compare to fully supervised approaches that train with action

Figure 4: Illustration of how the learned representation can be used for **a)** action-conditioned prediction by inferring the latent variable, $z_t$, from the action, and **b)** visual servoing by solving the control problem in latent space through iterated rollouts and then mapping the latent variable to robot control actions, $u_t$.

annotations: our implementations are based on Oh et al. (2015) for the reacher dataset and the more complex Finn & Levine (2017) for the BAIR dataset. For planning, we also compare to a model based on the approach of Agrawal et al. (2016) that learns the forward and inverse dynamics with direct supervision.

**Metrics.** In case of the action-conditioned video prediction we use the absolute angular position (obtained using a simple edge detection algorithm, see Appendix D) for the reacher dataset and the change of end effector position (obtained via manual annotation) for the BAIR dataset. We choose these metrics as they capture the direct consequences of applied actions, as opposed to more commonly used visual appearance metrics like PSNR or SSIM. For visual servoing in the reacher environment we measure the angular distance to the goal state at the end of servoing.

## 4.1 LEARNED STRUCTURE OF THE ACTION REPRESENTATIONS

First, we inspect the structure of the learned action space for our model. To do so, we train CLASP on the reacher dataset and visualize the learned representation. In Fig. 3, we show two-dimensional

---

[4]The original dataset provides two additional discrete actions: gripper closing and lifting. However, we found that, in this dataset, the spatial position in the horizontal plane explains most of the variance in the end effector position and therefore ignore the discrete actions in this work.

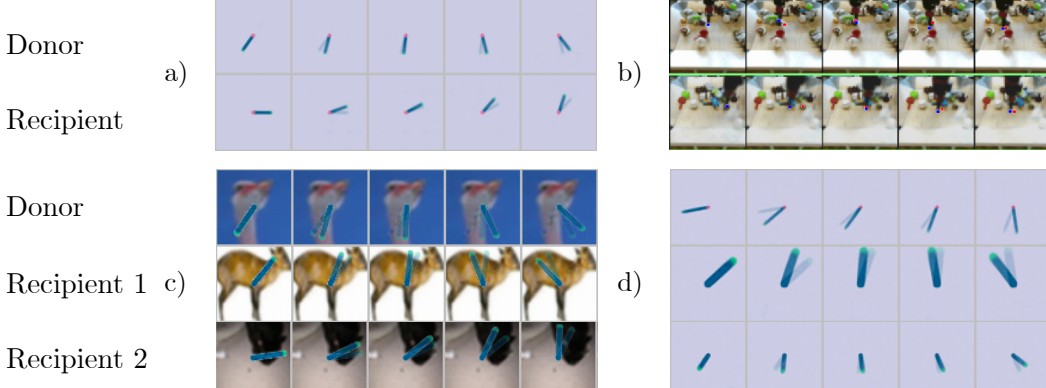

Figure 5: Transplantation of action representations $z$ from one sequence to another. We infer action representations from the donor sequence and use them to create the recipient sequences from a different initial state. **a)** the reacher dataset. The previous frame is superimposed onto each frame to illustrate the movement. **b)** the BAIR dataset. The previous and the current position of the end effector are annotated in each frame (red and blue dots, respectively) to illustrate the movement. **c)** reacher with varying backgrounds. **d)** reacher with varying agent shape. The synchronization of movement in the sequences suggests that the learned action representation is disentangled from static content. Best viewed on a screen. Additional generated videos are available at: `https://daniilidis-group.github.io/learned_action_spaces/`.

projections of samples, $z$, from the inference network, $q$, colored by the corresponding ground truth action, $u$. To find the two-dimensional subspace with maximal variability, we conducted Principal Component Analysis (PCA) on the means of the distributions generated by $q$. The first PCA dimension captures 99% of the variance, which is explained by the fact that the robot in consideration has one degree of freedom. While the baseline without composability training fails to learn a representation disentangled from the static content, our method correctly recovers the structure of possible actions of the robot.

## 4.2    ACTION-CONDITIONED VIDEO PREDICTION

We further verify that our model recovers a representation of actions and show that this allows us to use the model for two tasks. First, we show that it is possible to *transplant* the action representations $z$ from a given sequence into one with a different initial state. We run the approximate inference network $\text{MLP}_{\text{infer}}$ on the donor sequence to get the corresponding action representation $z$. We then use this sequence of representations $z$ together with a different conditioning image sequence to produce the recipient sequence. While the content of the scene changes, the executed actions should remain the same. Second, we show how our model can be used for *action-conditioned* video prediction. Given a ground truth sequence annotated with

Table 1: Action-conditioned video prediction results (mean $\pm$ standard deviation across predicted sequences). The "supervised" baseline is taken from Oh et al. (2015) for the reacher dataset and Finn & Levine (2017) for BAIR.

|  | Reacher | BAIR |
|---|---|---|
| Method | Abs. Error [in deg] | Rel. Error [in px] |
| Start State | $90.1 \pm 51.8$ | - |
| Random | $26.6 \pm 21.5$ | - |
| Denton & Fergus | $22.6 \pm 17.7$ | $3.6 \pm 4.0$ |
| CLASP (Ours) | $\mathbf{2.9 \pm 2.1}$ | $\mathbf{3.0 \pm 2.1}$ |
| Supervised | $2.6 \pm 1.8$ | $2.0 \pm 1.3$ |

actions $u$, we infer the representations $z$ directly from $u$ using $\text{MLP}_{\text{lat}}$. The inferred representations are fed into the generative model $p_\theta$ and the resulting sequences are compared to the original sequence.

The quantitative results in Table 1 show that the model trained with the composability objective on the reacher dataset successfully performs the task, with performance similar to the fully supervised model. Denton & Fergus (2018) performs the task only slightly better than random guessing. This shows that it is infeasible to infer the latent $z_t$ learned by the baseline model given only the action $u_t$,

and confirms our intuition about this from Fig. 3. The qualitative results in Fig. 5 (additional results in Figs. 12, 13 and 14 in the Appendix and on the website) further support this conclusion.

On the BAIR dataset, our model performs better than the baseline of Denton & Fergus (2018), reducing the difference between the best unsupervised method and the supervised baseline by 30 %. This is reflected in qualitative results as frames generated by the baseline model often contain artifacts such as blurriness when the arm is moving or ghosting effects with two arms present in the scene (Figs. 13 and 14 in the Appendix, videos on the website). These results demonstrate the promise of our approach in settings involving more complex, real-world interactions.

### 4.3 PLANNING IN THE LEARNED ACTION SPACE

Similarly to the true action space $u$, we can use the learned action space $z$ for planning. We demonstrate this on a visual servoing task. The objective of visual servoing is to move an agent from a start state to a goal state, given by images $x_0$ and $x_{\text{goal}}$, respectively. We use a planning algorithm similar to that of Finn & Levine (2017), but plan trajectories in the latent space $z$ instead of true actions $u$. We use MLP$_{\text{act}}$ to retrieve the actions that correspond to a planned trajectory.

Our planning algorithm, based on Model Predictive Control (MPC), is described in Appendix C. The controller plans by sampling a number of action trajectories and iteratively refining them with the Cross Entropy Method (CEM, Rubinstein & Kroese (2004)). The state trajectories are estimated by using the learned predictive model. We select the trajectory whose final state is closest to the goal and execute its first action. The distance between the states is measured using the cosine distance between VGG16 representations (Simonyan & Zisserman (2015)). Servoing terminates once the goal is reached or the maximum steps are executed. The baseline of Agrawal et al. (2016) uses a different procedure, as described in the original paper.

Table 2: Visual servoing performance measured as distance to the goal at the end of servoing (mean $\pm$ standard deviation).

| Reacher | |
|---|---|
| Method | Distance [deg] |
| Start Position | $97.8 \pm 23.7$ |
| Random | $27.0 \pm 26.8$ |
| Denton & Fergus (2018) | $14.1 \pm 10.7$ |
| CLASP (Ours) | $\mathbf{1.6 \pm 1.0}$ |
| Agrawal et al. (2016) | $2.0 \pm 1.5$ |
| Oh et al. (2015) | $1.8 \pm 1.5$ |
| CLASP (varied background) | $3.0 \pm 2.2$ |
| CLASP (varied agents) | $2.8 \pm 2.9$ |

We show qualitative results of a servoing rollout in the reacher environmet in Fig. 6 (left) and quantitative results in Table 2. The agent not only reaches the target but also plans accurate trajectories at each intermediate time step. The trajectory planned in the learned space can be correctly decoded into actions, $u$.

### 4.4 DATA EFFICIENCY

To validate the benefits of learning from passive observations, we measure the data efficiency of CLASP in Fig. 6 (right). In this setup, we train the methods on a large dataset of passive observations and a varied number of observations labeled with actions (100, 1000, 10000 videos). The supervised baselines, which cannot leverage pre-training with passive observations perform poorly in the low-data regime. In contrast, our model only needs a small number of action-labeled training sequences to achieve good performance, as it learns the structure of actions from passive observations. In the abundant data regime, our model still performs on par with both supervised baselines. We observed similar results for action-conditioned prediction experiments, summarized in Table 3 in the Appendix. These results suggest that our planning approach can be used when the action-labeled data are limited.

### 4.5 ROBUSTNESS TO VARYING VISUAL CHARACTERISTICS

To test the robustness of our approach to different kinds of visual variability in the environment, we conduct experiments on two versions of the reacher dataset with additional variability. In the first, the background of each sequence is replaced with a randomly drawn CIFAR-10 image (Krizhevsky (2009)). In the second, we vary the width and length of the reacher arm in each sequence. We test models trained on these datasets on sequences with variations not seen during training but drawn from the same distribution. The experimental setup is described in more detail in Appendix E.

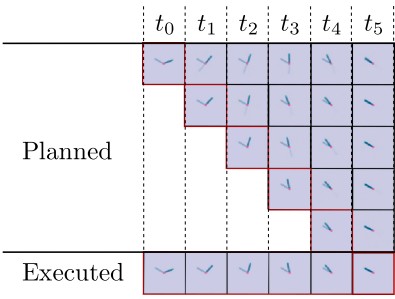 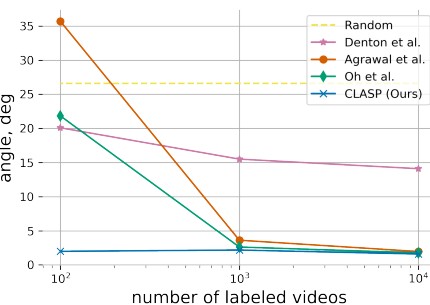

Figure 6: Visual servoing on the reacher task. **Left**: Planned and executed servoing trajectories. Each of the first five rows shows the trajectory re-planned at the corresponding timestep. The first image of each sequence is the current state of the system, and the images to the right of it show the model prediction with the lowest associated cost. The target state (the reacher pointing to the upper left) is shown superimposed over each image. **Right**: Data efficiency measured as final distance to the goal after servoing, shown depending on the number of videos used in training. Each point represents a model trained on a dataset with a restricted number of action-annotated training sequences. Full results are in Table 4 in the appendix.

As shown in Table 2, our model can reliably discover the agent's action space and perform visual servoing under increased visual variability. The transplantation sequences in Fig. 5 show that the action semantics are preserved across changes to the appearance of the environment that do not alter the dynamics. This is evidence that the learned representation captures the dynamics of the environment and is not sensitive to changes in visual characteristics that do not affect the agent's action space. In these two settings, CLASP also requires orders of magnitude less action-conditioned data than the supervised baselines (see Table 4 in the appendix). Our results, combined with the data efficiency result, suggest that our method is robust to visual changes and can be used for passive learning from videos that are obtained under different visual conditions, or even videos of different agents, such as videos obtained from the Internet, as long as the action space of the observed agents coincides with the target agent.

## 5 CONCLUSION

We have shown a way of learning the structure of an agent's action space from visual observations alone by imposing the properties of minimality and composability on a latent variable for stochastic video prediction. This strategy offers a data-efficient alternative to approaches that rely on fully supervised action-conditioned methods. The resulting representation can be used for a range of tasks, such as action-conditioned video prediction and planning in the learned latent action space. The representation is insensitive to the static scene content and visual characteristics of the environment. It captures meaningful structure in synthetic settings and achieves promising results in realistic visual settings.

ACKNOWLEDGEMENTS

We thank Nikos Kolotouros and Karl Schmeckpeper for help with annotation, Kenneth Chaney and Nikos Kolotouros for computing support, Stephen Phillips and Nikos Kolotouros for helpful comments on the document, and the members of the GRASP laboratory and CLVR laboratory for many fruitful discussions. We also thank the audiences of the 2018 R:SS workshop on Learning and Inference in Robotics, 2018 International Computer Vision Summer School, and 2018 NeurIPS workshop on Probabilistic Reinforcement Learning and Structured Control for useful feedback. We are grateful for support through the following grants: NSF-DGE-0966142 (IGERT), NSF-IIP-1439681 (I/UCRC), NSF-IIS-1426840, NSF-IIS-1703319, NSF MRI 1626008, ARL RCTA W911NF-10-2-0016, ONR N00014-17-1-2093, and by Honda Research Institute. K.G.D. is supported by a Canadian NSERC Discovery grant. K.G.D. contributed to this article in his personal capacity as an Associate Professor at Ryerson University.

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

## A    STOCHASTIC VIDEO PREDICTION

We use an architecture similar to SVG-FP of Denton & Fergus (2018). Input images $x_t$ are encoded using a convolutional neural network $\text{CNN}_e(\cdot)$ to produce a low-dimensional representation $\text{CNN}_e(x_t)$; output image encodings can be decoded with a neural network with transposed convolutions $\text{CNN}_d(\cdot)$. We use a Long Short-Term Memory network $\text{LSTM}(\cdot, \cdot)$ for the generative network $\mu_\theta(x_{t-1}, z_t) = \text{CNN}_d(\text{LSTM}(\text{CNN}_e(x_{t-1}), z_t))$, and a multilayer perceptron $\text{MLP}_{\text{infer}}$ for the approximate inference network $[\mu_\phi(x_t, x_{t-1}), \sigma_\phi(x_t, x_{t-1})] = \text{MLP}_{\text{infer}}(\text{CNN}_e(x_t), \text{CNN}_e(x_{t-1}))$.

During training, our model first observes $K$ past input frames. From these observations, the model generates $K - 1$ corresponding latents $z_{2:K}$ and predicts $K - 1$ images $\hat{\mathbf{x}}_{2:K} = \mu_\theta(\mathbf{x}_{1:K-1}, \mathbf{z}_{2:K})$. The model generates $T - K$ further future images: $\hat{\mathbf{x}}_{K+1:T} = \mu_\theta(\hat{\mathbf{x}}_{1:T-1}, \mathbf{z}_{1:T})$. At test time, latents $z_t$ are sampled from the prior $\mathcal{N}(0, I)$, and the model behaves identically otherwise. We show samples from the stochastic video prediction model in Fig. 11.

Unlike in Denton & Fergus (2018), the generating network $p_\theta$ does not observe ground truth frames $\mathbf{x}_{K+1:T-1}$ in the future during training but autoregressively takes its own predicted frames $\hat{\mathbf{x}}_{K+1:T-1}$ as inputs. This allows the network LSTM to generalize to observing the generated frame encodings $\text{LSTM}(\text{CNN}_e(x_{t-1}), z_t)$ at test time when no ground truth future frames are available. We use a recurrence relation of the form $\text{LSTM}(\text{LSTM}(x_{t-2}, z_{t-1}), z_t)$. To overcome the generalization problem, Denton & Fergus (2018) instead re-encode the produced frames with a recurrence relation of the form $\text{LSTM}(\text{CNN}_e(\text{CNN}_d(\text{LSTM}(x_{t-2}, z_{t-1}))), z_t)$. Our approach omits the re-encoding, which saves a considerable amount of computation.

## B    EXPERIMENTAL PARAMETERS

For all experiments, we condition our model on five images and roll out ten future images. We use images with a resolution of $64 \times 64$ pixels. The dimension of the image representation is $\dim(g(x)) = 128$, and the dimensions of the learned representation are $\dim(z) = \dim(\nu) = 10$. For the reacher dataset, we use the same architecture as Denton & Fergus (2018) for the $f, g$ and LSTM networks. For experiments with the standard blue background (i.e. all except the varied background experiment) we do not use temporal skip-connections. For the BAIR dataset, we do not use $f, g$ and use the same model as Lee et al. (2018) for LSTM. The $\text{MLP}_{\text{infer}}$ has two hidden layers with 256 and 128 units, respectively. The $\text{MLP}_{\text{comp}}, \text{MLP}_{\text{lat}},$ and $\text{MLP}_{\text{act}}$ networks each have two hidden layers with 32 units. For $\text{MLP}_{\text{lat}}$ and $\text{MLP}_{\text{act}}$, we tried wider and deeper architectures, but this did not seem to improve performance of either our method or the baseline without composability. This is probably because the latent space in our experiments had either a simple representation that did not need a more powerful network to interpret it, or was entangled with static content, in which case even a more powerful network could not learn the bijection. The number of latent samples $z$ used to produce a trajectory representation $\nu$ is $C = 4$. For all datasets, $\beta_z = 10^{-2}, \beta_\nu = 10^{-8}$ We use the leaky ReLU activation function in the $g, f,$ and MLP networks. We optimize the objective function using the Adam optimizer with parameters $\beta_1 = 0.9, \beta_2 = 0.999$ and a learning rate of $2 \times 10^{-4}$. All experiments were conducted on a single high-end NVIDIA GPU. We trained the models for 4 hours on the reacher dataset, for one day on the BAIR dataset.

We found the following rule for choosing both bottleneck parameters $\beta_z$ and $\beta_\nu$ to be both intuitive and effective in practice: they should be set to the highest value at which samples from the approximate inference $q$ produce high-quality images. If the value is too high, the latent samples will not contain enough information to specify the next image. If the value is too low, the divergence between the approximate inference and the prior will be too large and therefore the samples from the prior will be of inferior quality. We note that the problem of determining $\beta$ is not unique to this work and occurs in all stochastic video prediction methods, as well as VIB and $\beta$-VAE.

## C    VISUAL SERVOING

We use Algorithm 1 for visual servoing. At each time step, we initially sample $M$ latent sequences $z_0$ from the prior $\mathcal{N}(0, I)$ and use the video prediction model to retrieve $M$ corresponding image sequences $\tau$, each with $K$ frames. We define the cost of an image trajectory as the cosine distance between the VGG16 (Simonyan & Zisserman (2015)) feature representations of the target image and

Table 3: Average absolute angle error (mean $\pm$ standard deviation) for action-conditioned video prediction. Note that we could not detect angles on some sequences for the action-conditioned baseline of Oh et al. (2015) trained on only 100 sequences due to bad prediction quality.

| Reacher | | | |
|---|---|---|---|
| Method | Angle Error [deg] | | |
| Training Sequences | 100 | 1000 | 10 000 |
| Start Position | ——————— $90.6 \pm 52.0$ ——————— | | |
| Random | ——————— $27.7 \pm 22.2$ ——————— | | |
| Denton & Fergus (2018) | $27.6 \pm 22.8$ | $23.8 \pm 18.6$ | $23.6 \pm 18.3$ |
| CLASP (Ours) | $\mathbf{2.9 \pm 2.0}$ | $\mathbf{2.9 \pm 2.0}$ | $3.0 \pm 2.0$ |
| Oh et al. (2015) | - | $5.6 \pm 4.5$ | $\mathbf{2.7 \pm 1.9}$ |

Table 4: Visual servoing performance and data efficiency.

| Reacher | | | |
|---|---|---|---|
| Method | Distance [deg] | | |
| Training Sequences | 100 | 1000 | 10 000 |
| Start Position | ——————— $97.8 \pm 23.7$ ——————— | | |
| Random | ——————— $27.0 \pm 26.8$ ——————— | | |
| Denton & Fergus (2018) | $20.9 \pm 13.0$ | $15.5 \pm 13.1$ | $14.1 \pm 10.7$ |
| CLASP (Ours) | $\mathbf{2.0 \pm 2.2}$ | $\mathbf{2.2 \pm 1.8}$ | $\mathbf{1.6 \pm 1.0}$ |
| Agrawal et al. (2016) | $32.7 \pm 21.7$ | $3.6 \pm 3.1$ | $2.0 \pm 1.5$ |
| Oh et al. (2015) | $21.8 \pm 12.9$ | $2.6 \pm 2.6$ | $1.8 \pm 1.5$ |
| CLASP (varied background) | $1.5 \pm 1.3$ | $3.8 \pm 3.5$ | $3.0 \pm 2.2$ |
| CLASP (varied agents) | $2.0 \pm 1.0$ | $2.3 \pm 3.4$ | $2.8 \pm 2.9$ |

the final image of each trajectory. This is a perceptual distance, as in Johnson et al. (2016). In the update step of the Cross Entropy Method (CEM) algorithm, we rank the trajectories based on their cost and fit a diagonal Gaussian distribution to the latents $\boldsymbol{z}'$ that generated the $M'$ best sequences. We fit one Gaussian for each prediction time step $k \in K$. After sampling a new set of latents $\boldsymbol{z}_{n+1}$ from the fitted Gaussian distributions we repeat the procedure for a total of $N$ iterations.

Finally, we pick the latent sequence corresponding to the best rollout of the last iteration and map its first latent sample to the output control action using the learned mapping: $u^* = \text{MLP}_{\text{act}}(z^*_{N,0})$. This action is then executed in the environment. The action at the next time step is chosen using the same procedure with the next observation as input. The algorithm terminates when the specified number of servoing steps $T$ has been executed.

---

**Algorithm 1** Planning in the learned action space

---

**Require:** Video prediction model $\hat{x}_{t:t+K} = \mu_\theta(x_{1:t-1}, z_{2:t+K})$
**Require:** Start and goal images $i_0$ and $i_{\text{goal}}$
 1: **for** $t = 1 \ldots T$ **do**
 2: $\quad$ Initialize latents from prior: $\boldsymbol{z}_0 \sim \mathcal{N}(0, I)$
 3: $\quad$ **for** $n = 0 \ldots N$ **do**
 4: $\quad\quad$ Rollout prediction model for $K$ steps, obtain $M$ future sequences $\boldsymbol{\tau} = \hat{\boldsymbol{x}}_{t:t+K}$
 5: $\quad\quad$ Compute cosine distance between final and goal image: $c(\tau) = \cos(\hat{x}_{t+K}, i_{\text{goal}})$
 6: $\quad\quad$ Choose $M'$ best sequences, refit Gaussian distribution: $\boldsymbol{\mu}_{n+1}, \boldsymbol{\sigma}_{n+1} = \text{fit}(\boldsymbol{z}'_n)$
 7: $\quad\quad$ Sample new latents from updated distribution: $\boldsymbol{z}_{n+1} \sim \mathcal{N}(\boldsymbol{\mu}_{n+1}, \boldsymbol{\sigma}_{n+1})$
 8: $\quad$ **end for**
 9: $\quad$ Map first latent of best sequence to action: $u^* = \text{MLP}_{\text{act}}(z^*_{N,0})$
10: $\quad$ Execute $u^*$ and observe next image
11: **end for**

---

Table 5: Hyperparameters for the visual servoing experiments. We sample an angle uniformly from the angle difference range to create each subsequent image in a sequence.

| Servoing Parameters | |
|---|---|
| Servoing timesteps $(T)$ | 5 |
| Servoing horizon $(K)$ | 5 |
| # servoing sequences $(M)$ | 10 |
| # refit sequences $(M')$ | 3 |
| # refit iterations $(N)$ | 4 |
| Angle difference range | $[0°, 40°]$ |

The parameters used for our visual servoing experiments are listed in Tab. 5.

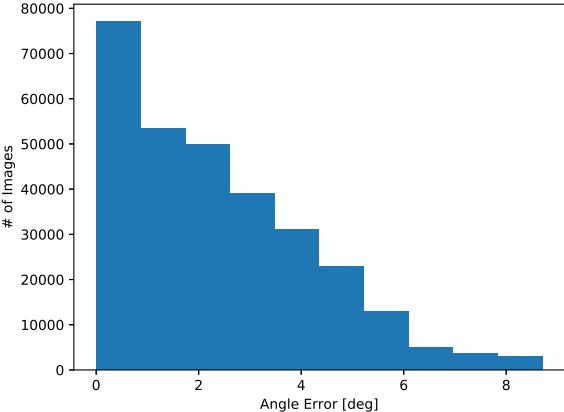

Figure 7: Error histogram of the angle detection algorithm on the reacher training set. The output of this algorithm is used as a form of surrogate ground truth to evaluate model performance.

## D  ANGLE DETECTION ALGORITHM

We employ a simple, hand-engineered algorithm to quickly retrieve the absolute angle values from the images of the reacher environment. First we convert the input to a grayscale image and run a simple edge detector to obtain a binary image of the reacher arm. We smooth out noise by morphological opening. We compute the Euclidean distance to the image center for all remaining non-zero pixels and locate the reacher tip at the pixel closest to the known reacher arm length. This gives us the absolute reacher arm angle.

To evaluate the accuracy of our angle detection algorithm, we estimated the angle for all images of the simulated training dataset and compare it to ground truth. A histogram of the angle errors of our algorithm is displayed in Fig. 7. All errors are below 10 degrees and the majority are smaller than 5 degrees. This suggests the output of this model is of a suitable quality to serve as surrogate ground truth. A second approach that used a neural network to regress the angle directly from the pixels achieved similar performance. We attribute the errors to the discretization effects at low image resolutions – it is impossible to achieve accuracy below a certain level due to the discretization.

## E  EXPERIMENTS WITH VARYING ENVIRONMENTS

### E.1  ROBUSTNESS TO CHANGING STATIC BACKGROUND

We test the robustness of our method to different static backgrounds by replacing the uniform blue background with images from the CIFAR-10 training set (Krizhevsky (2009)). For each sequence we sample a single background image that is constant over the course of the entire sequence. At test time we use background images that the model has not seen at training time, i.e. sampled from a held-out subset of the CIFAR-10 training set. As in previous experiments, we first train our model on pure visual observations without action-annotations. We then train the networks $MLP_{lat}$ and $MLP_{act}$ on a small set of action-annotated sequences to convergence. For the visual servoing we follow the same algorithm as in the previous experiments (see Appendix C).

Qualitative servoing results of our method on the dataset with varied backgrounds are shown in Fig. 9 and quantitative results in Figure 6 (right). The model accurately predicts the background image into the future and successfully discovers and controls the action space of the agent. The fact that the same bijective mapping between latents and actions works for all backgrounds suggests that the network is able to disentangle the static content of the scene and the dynamics attributed to the moving reacher arm. In addition, we show trajectory transplantation between different backgrounds in Fig. 8 (top), which further validates the claim that the learned latent represents the action consistently, independent of the background.

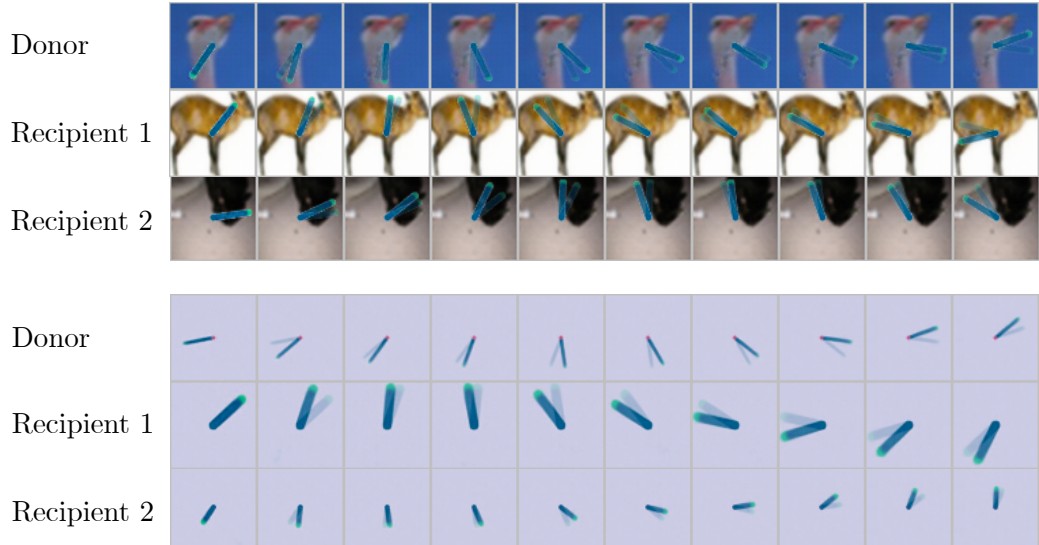

Figure 8: Trajectory transplantation with differing visual characteristics. The trajectory from the top sequence is transplanted to a different environment and initial state in each of the two bottom sequences. Our model achieves almost perfect accuracy, which validates that it has indeed learned a representation of actions disentangled from the static content, such as the background, agent's appearance, and the initial state. The previous frame is superimposed onto each frame to illustrate the movement. **Top:** dataset with varying backgrounds. **Bottom:** dataset with varying robots. Additional generated videos are available at: `https://daniilidis-group.github.io/learned_action_spaces/`.

### E.2 LEARNING FROM AGENTS WITH DIFFERENT VISUAL APPEARANCE

We test the ability of our method to learn from agents that differ in their visual appearance from the agent used at test time, but that share a common action space. We construct a dataset in which we vary parameters that determine the visual characteristics of the reacher arm, specifically its thickness and length (see Fig. 10, left). In total our training dataset comprises 72 different configurations spanning a wide variety of visual appearances.

We show a qualitative example of a servoing trajectory in Fig. 10 (right). We additionally evaluate the efficacy of training on the novel dataset by following the procedure employed in Section 4.4: we train the mapping between latent representation $z$ and actions to convergence on action-annotated subsets of the training data of varying sizes. The servoing errors in Figure 6 (right) show that we achieve comparable performance independent of whether we train on the target agent we test on or on a set of agents with different and varied visual appearances. Our model is able to learn a latent representation that captures the action space shared between all the agents seen at training time. We can then learn the mapping between this abstract action space and the actions of the agent with the novel visual appearance from a small number of action-annotated sequences. In addition, we show trajectory transplantation between different agents in Fig. 8 (bottom) that further validates our claim that the learned latent represents the action consistently, independent of the agent.

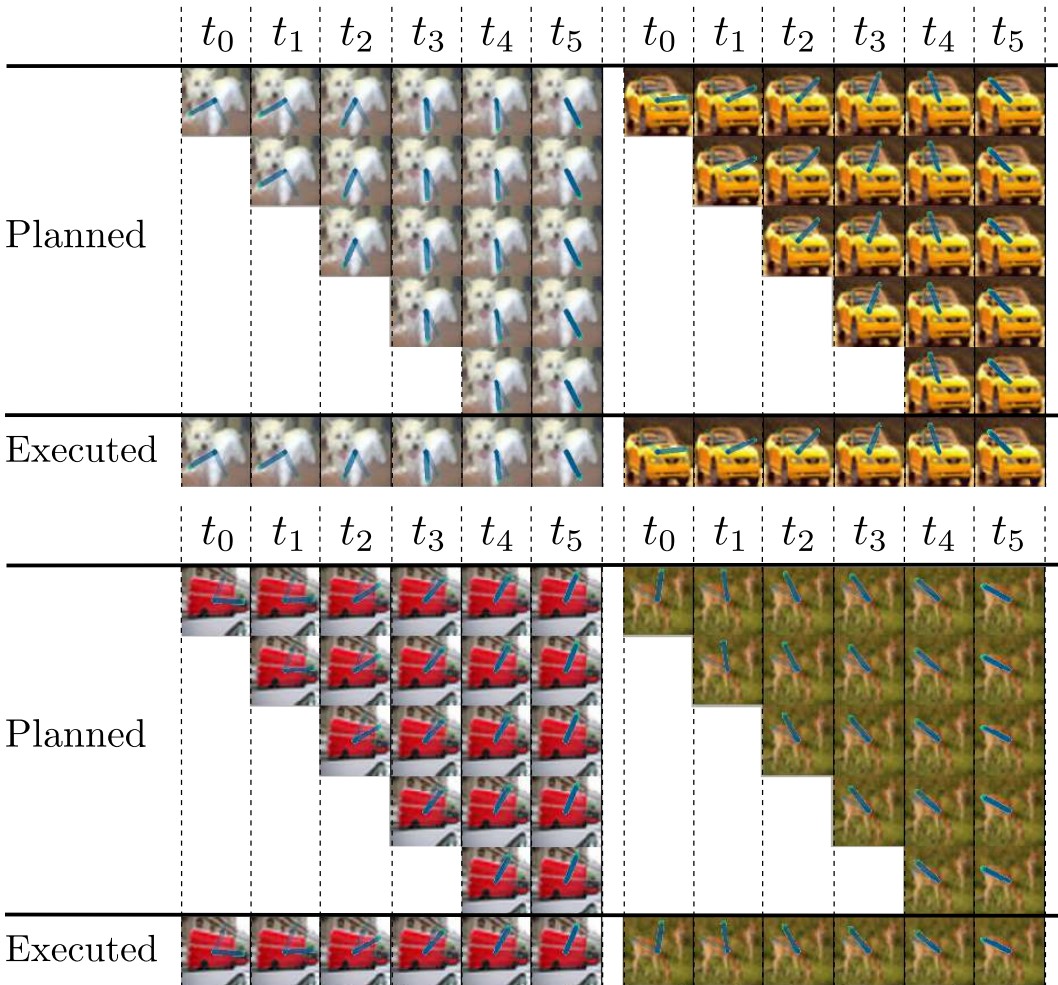

Figure 9: Servoing examples with randomly sampled static CIFAR-10 backgrounds. The figure layout follows the layout of Fig. 6 (left).

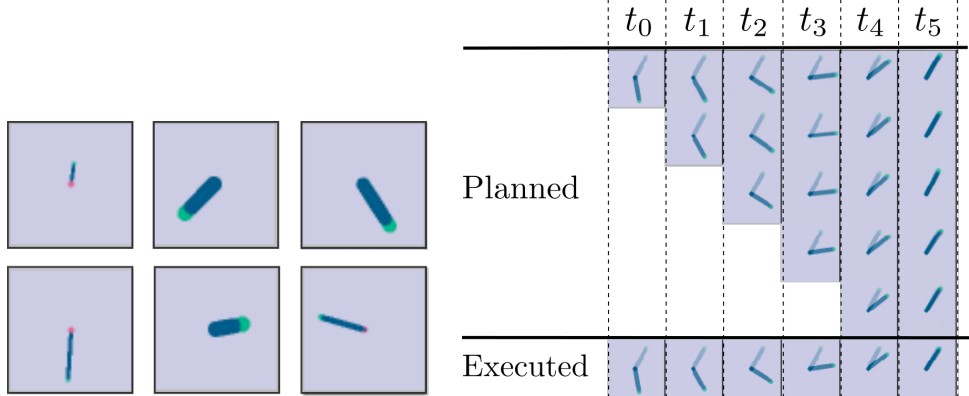

Figure 10: Learning from agents with varied visual appearance. **Left**: Sample agent configurations from the training set. We cover a variety of visual appearances (i.e. arm lengths and widths) but not the configuration used for testing. **Right**: Test time servoing example after pre-training on observations of agents with varied visual appearances. The figure layout follows the layout of Fig. 6 (left).

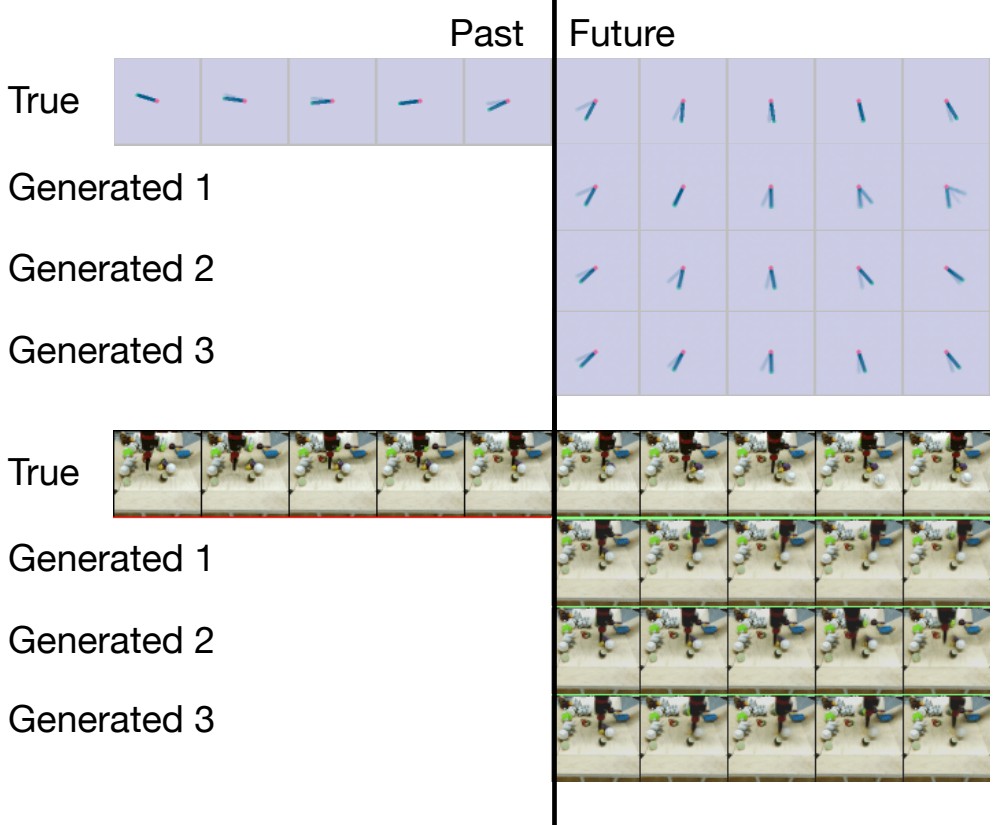

Figure 11: Typical sequences sampled from the stochastic video prediction model. In the past, the samples $z$ are generated from the approximate inference distribution and match the ground truth exactly. In the future, $z$ is sampled from the prior, and correspond to various possible futures. These three sequences are different plausible continuations of the same past sequence. This shows that the model is capable of capturing the stochasticity of the data. Only five of ten predicted frames are shown for clarity. Additional generated videos are available at: `https://daniilidis-group.github.io/learned_action_spaces/`.

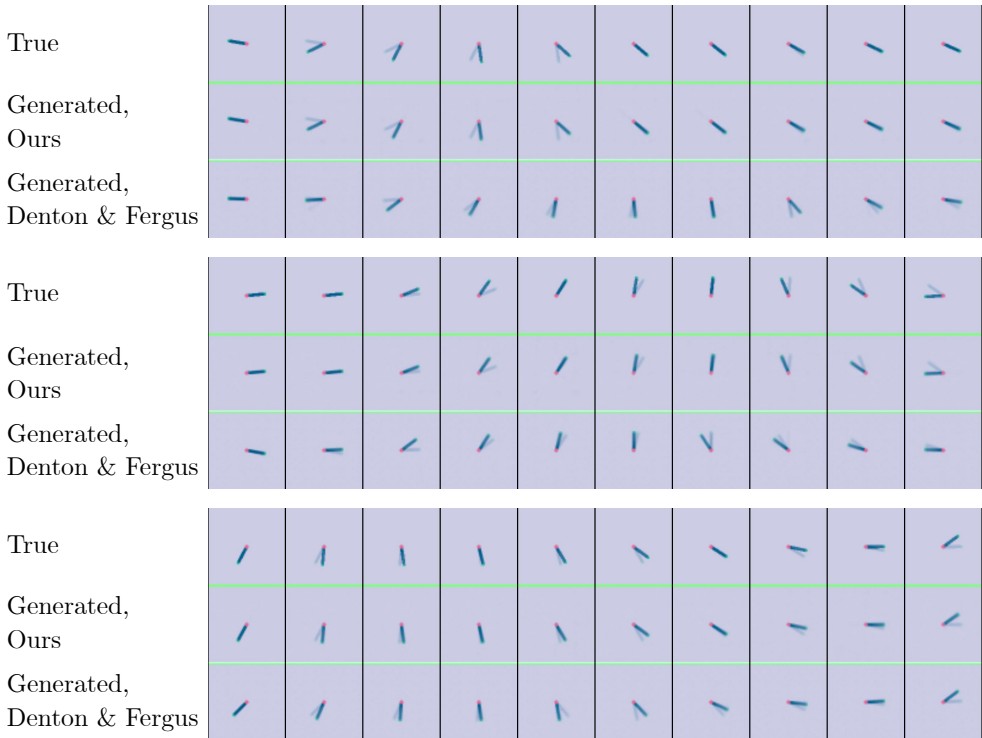

Figure 12: Typical action-conditioned prediction sequences on the reacher dataset. Each example shows **top**: the ground truth sequence, **middle**: our predictions, **bottom:** predictions of the baseline model (Denton & Fergus (2018)). To illustrate the motion, we overlay the previous position of the arm in each image (transparent arm). Our method produces sequences that are perfectly aligned with the ground truth. The baseline never matches the ground truth motion and is only slightly better than executing random actions. Best viewed on a computer, additional generated videos are available at: https://daniilidis-group.github.io/learned_action_spaces/.

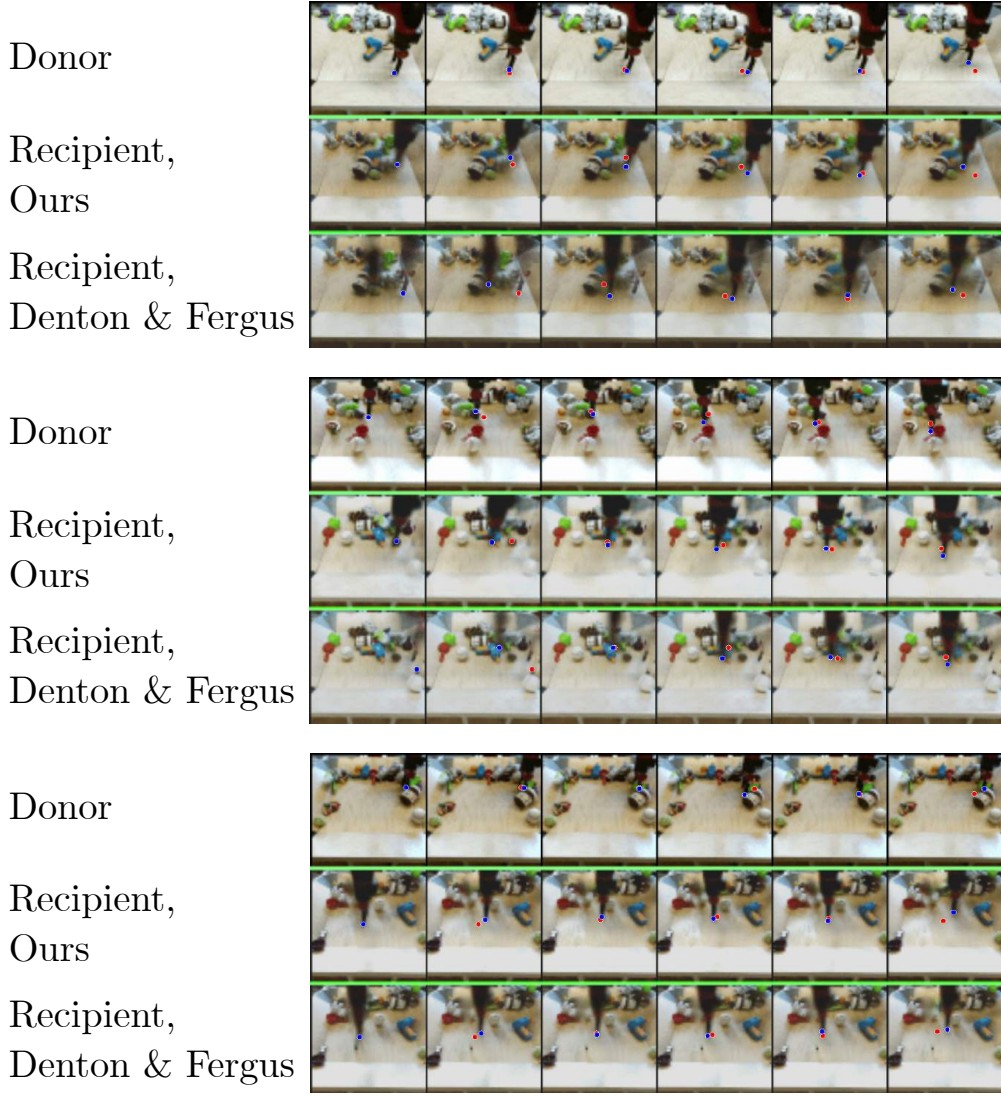

Figure 13: Failure cases of the baseline model on trajectory transplantation. Each example shows **top**: the ground truth sequence, **middle**: our predictions, **bottom:** predictions of the baseline model (Denton & Fergus (2018)). The position of the end effector at the current (blue) and previous (red) timestep is annotated in each frame. The baseline often produces images with two different robot arms and other artifacts. Only six of ten predicted frames are shown for clarity. Best viewed on a computer, additional generated videos are available at: `https://daniilidis-group.github.io/learned_action_spaces/`.

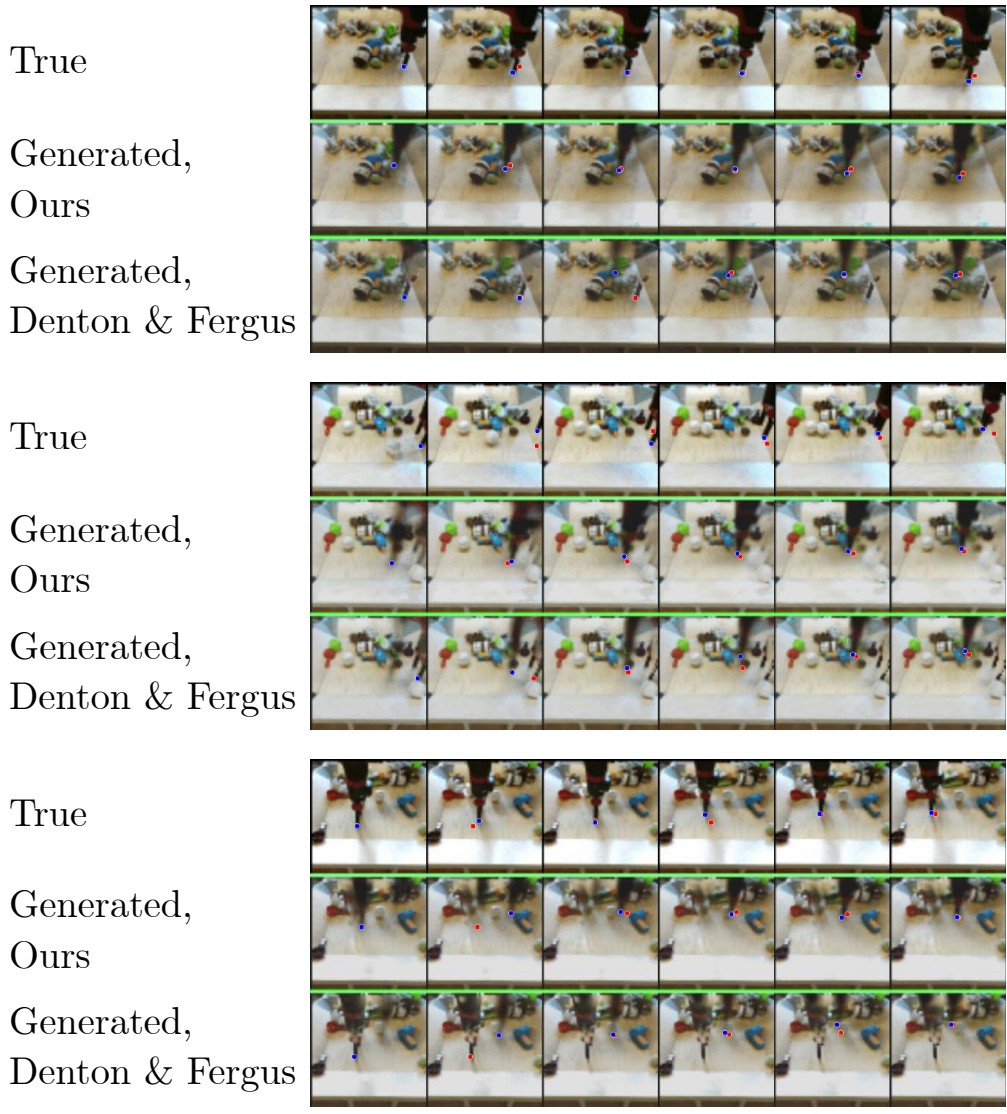

Figure 14: Baseline failure cases on action-conditioned video prediction. Each example shows **top**: ground truth sequence, **middle**: our predictions, **bottom:** Denton & Fergus (2018) baseline. The previous and the current position of the end effector are annotated in each frame. The baseline often produces images with two different robot arms and other artifacts. Only six of ten predicted frames are shown for clarity. Best viewed on a computer, additional generated videos are available at: `https://daniilidis-group.github.io/learned_action_spaces/`.

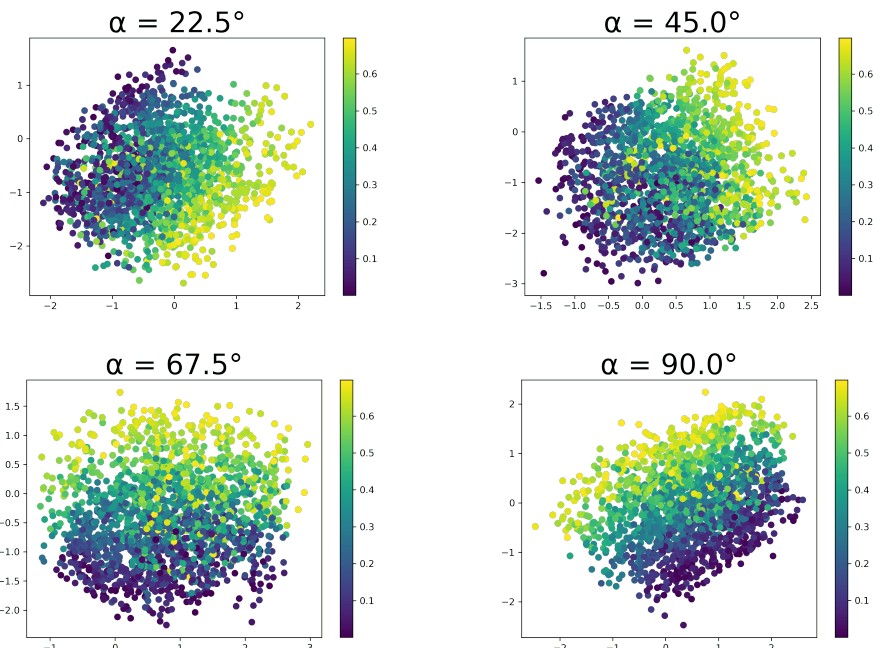

Figure 15: Visualization of the structure of the learned latent space of the baseline model without composability training on the reacher dataset. The visualization is done in the same manner as in Fig. 3. Here, action representations $z_t$ are shown as a function of the absolute angle ($\alpha$) of the reacher arm at time $t-1$ and the relative angle between the reacher at time $t$ and $t-1$. We see that the encoding of action learned by the baseline is entangled with the absolute position of the reacher arm. While this representation can be used to predict the consequences of actions given the previous frame, it is impossible to establish a bijection between $u_t$ and $z_t$ as the correspondence depends on the previous frame $x_{t-1}$. Moreover, it is impossible to compose two samples of such a $z$ without access to the intermediate frame. This representation is *minimal*, as it is a linear transformation (a rotation) of the known optimal representation $u_t$ (the ground truth actions). This suggests that composability plays an important role in learning a disentangled representation of actions.

