# OpenReview forum: "Learning what you can do before doing anything"
_ICLR.cc/2019/Conference_

### Official Review · AnonReviewer1 · 2018-11-03
**Good paper, needs some clarification in the experimental section and the introduction**

**Rating:** 6
**Confidence:** 3

**Review:**

PAPER SUMMARY
-------------
This paper proposes an approach to video prediction which autonomously finds an action space encoding differences between subsequent frames. This approach can be used for action-conditioned video prediction and visual servoing.
Unlike related work, the proposed method is initially trained on video sequences without ground-truth actions. A representation for the action at each time step is inferred in an unsupervised manner. This is achieved by imposing that the representation of this action be as small as possible, while also being composable, i.e. that that several actions can be composed to predict several frames ahead.
Once such a representation is found, a bijective mapping to ground truth actions can be found using only few action-annotated samples. Therefore the proposed approach needs much less annotated data than approaches which directly learn a prediction model using actions and images as inputs.

The approach is evaluated on action-conditioned video prediction and visual servoing. The paper shows that the learned action-space is meaningful in the sense that applying the same action in different initial condition indeed changes the scenes in the same manner, as one would intuitively expect. Furthermore, the paper shows that the approach achieves state of the art results on a action-conditioned video prediction dataset and on a visual servoing task.

POSITIVE POINTS
---------------
The idea of inferring the action space from unlabelled videos is very interesting and relevant.

The paper is well written.

The experimental results are very interesting, it is impressive that the proposed approach manages to learn meaningful actions in an unsupervised manner (see e.g. Figure 3).

NEGATIVE POINTS
---------------
It is not exactly clear to me how the model is trained for the quantitative evaluation. On which sequences is the bijective mapping between inferred actions and true actions learned? Is is a subset of the training set? If yes, how many sequences are used? Or is this mapping directly learned on the test set? This, however, would be an unfair comparison in my opinion, since then the actions would be optimized in order to correctly predict on the tested sequences.

The abstract and introduction are too vague and general. It only becomes clear in the technical and experimental section what problem is addressed in this paper.

---

> ### Author Response · Authors · 2018-11-09
> **Added details of training setup, improved abstract and introduction**
>
> We thank the reviewer for the helpful comments. We have updated the manuscript to address the points raised, and we believe the paper is now clearer. We address the reviewer's questions and comments as follows:
>
> 1. Q: Which sequences are used to train the bijective mapping between the learned action representation and the true action labels?
> A: The bijective mapping is trained using a small subset of the training data that is annotated with action labels. We do not use any test sequences for training. We have added a sentence clarifying this issue to Section 3.3 of the paper: “To determine this correspondence, we learn a simple bijective mapping from a small number of action-annotated frame sequences from the training data.”
>
> 2. Q: How many sequences are used to train the bijective mapping?
> A: For visual servoing, we use 100, 1000, or 10k action-annotated training sequences to train the bijective mapping, as shown in Figure 5 in the appendix. Our results show that 100 sequences are sufficient to train the bijective mapping to convergence. The supervised baseline needs up to 10k action-annotated sequences to converge to the same level of performance.
>
> During our experiments on action-conditioned video prediction, we also observed that 100 sequences were sufficient for our method to converge. The original submission did not include results showing prediction performance as a function of the number of action-annotated sequences used, so we have added an experiment to test this specifically. This experiment confirms that, as for visual servoing, our method converges with as few as 100 examples, while the baseline supervised method requires up to 10k examples. These results are shown in Table 1 in the appendix. We have updated the text in Section 4.1 of the paper to reflect this.
>
> 3. Q: The abstract and introduction are too vague and general
> A: We have updated the abstract and the 3rd and the 4rd paragraph of the introduction to include more detail about the problem addressed in the paper and the method we use to approach it. We believe the problem addressed in our paper - learning a representation that disentangles an agent's action space from unlabeled video - can now be much more clearly understood from the abstract and introduction. We have also changed these sections to emphasize that we add a new loss term that enforces composability and that our model learns a mapping between the latent action representation and the true action labels from a small number of action-annotated sequences. We emphasize that this approach enables (i) action-conditioned video prediction and (ii) trajectory planning for servoing in latent space, without requiring a large, action-annotated dataset.
>
> If you still feel that there are issues with the manuscript that would prevent you from raising your score, please point these out so that we can address them.
>
>
> ------
>
> We updated this response to reflect the numbering of figures in the revised manuscript

---

### Official Review · AnonReviewer2 · 2018-11-06
**I like the idea, but concerns with experimental evaluation**

**Rating:** 6
**Confidence:** 4

**Review:**

The authors propose a way to learn models that predict what will happen next in scenarios where action-labels are not available in abundance. The agents extend previous work by proposing a compositional latent-variable model. Results are shown on BAIR (robot pushing objects) and simulated reacher datasets. The results indicate that it is possible to learn a bijective mapping between the latent variables inferred from a pair of images and the action executed between the observations of the two images.

I like the proposed model and the fact that it is possible to learn a bijection between the latent variables and actions is cute. I have following questions/comments:

(a) The authors have to learn a predictive model from passive data (i.e. without having access to actions). Such models are useful, if for example an agent can observe other agents or internet videos and learn from them. In such scenarios, while it would be possible to learn “a” model using the proposed method, it is unclear how the bijective mapping would be learnt, which would enable the agent to actually use the model to perform a task that it is provided with.
In the current setup, the source domain of passive learning and target domain from which action-labelled data is available are the same. In such setups, the scarcity of action-labelled data is not a real concern. When an agent acts, it trivially has access to its own actions. So collecting observation, action trajectories is a completely self-supervised process without requiring any external supervision.

(b)  How is the model of Agrawal 2016 used for visual serving? Does it used the forward model in the feature space of the inverse model or something else?

(c) In the current method, a neural network is used for composition. How much worse would a model perform if we simply compose by adding the feature vectors instead of using a neural network. It seems like a reasonable baseline to me. Also, how critical is including binary indicator for v/z in the compositional model?

Overall, I like the technical contribution of the paper. The authors have a very nice introduction on how humans learn from passive data. However, the experiments make a critical assumption that domains that are used for passive and action-based learning are exactly the same. In such scenarios, action-labeled data is abundantly available. I would love to see some results and/or hear the authors thoughts on how their method can be used to learn by observing a different agent/domain and transfer the model to act in the agent’s current domain. I am inclined to vote for accepting the paper if authors provide a convincing rebuttal.

---

> ### Author Response · Authors · 2018-11-14
> **Added experiment with different source and target agents, clarified technical questions**
>
> We thank the reviewer for the helpful comments and suggestions. We have updated the manuscript to address all points raised. In particular, the suggested experiment shows the applicability of our method to cases where training and test agents differ visually but share the same action space. We address the reviewer's questions and comments as follows:
>
> 1. Q: Can the proposed model be applied on different source and target domains?
> A: As suggested by the reviewer, we ran an additional experiment to probe whether our model could learn from observations in a domain different from the target domain. The experimental results are now described in Section E.2 of the appendix. We train the model on a modified reacher environment where the training data consists of visual observations of reacher agents with varying arm lengths and widths. Importantly, our model does not have access to labeled actions for any training agents. We test the network performance on an agent with an arm length/width configuration that was never observed at training time but that shares the action space of the training agents. We use the target agent to collect the small dataset of action-annotated sequences needed to learn the bijective mapping. We show that the representation learned by our network successfully generalizes to the target agent in both visual servoing and transplantation experiments. We show that our network needs as few as 100 video sequences from the target domain to learn the correct bijection, even when trained only on agents with different properties than the target agent, while the supervised baselines need up to 10k sequences, see Figure 5 (right). Along with results suggesting our model learns to accommodate background variations (see Section E.1 of the Appendix and our response to Reviewer 3), this suggests the potential of our method to learn from purely visual data obtained from, e.g., Internet videos and be applied to control a different agent.
>
> 2. Q: How is the model of Agrawal et al. 2016 used for servoing? Does it use the inverse model?
> A: We use the model of Agrawal et al. 2016 for servoing exactly as proposed in the original paper, which requires the inverse model. Given the the target image and the input image sequence up to the current image, the trained inverse dynamics model outputs an action which is executed for servoing. At the next time step, the resulting observation is fed back into the model as the initial image and the process is repeated. The forward model of Agrawal et al. 2016 produces only state embeddings (not images), so it cannot be used for servoing by itself. We have clarified this point in the second paragraph of Section 4.2 of the manuscript.
>
> 3. Q: How does an additive composition baseline perform?
> A: In our experiments, an additive baseline did not produce meaningful results on action-conditioned prediction or other measures of disentanglement. This might be because the actions considered in the reacher experiment in fact do not form a group isomorphic to any additive vector space (here, actions are elements of the circle group); that is, this action space cannot be modeled with vector addition. It might be possible to get around this issue by using addition modulo some number instead of naive addition, but it is not clear how this number should be chosen. Using a neural network as the composition operator ensures that any group can be efficiently approximated with this neural network.
>
> 4. How critical is including binary indicator for v/z in the compositional model?
> A: To train the compositional loss, we require a way of decoding trajectory representations (together with a sequence of previous frames) into images. One way to do this would be to include an additional LSTM that learns to operate on trajectory representations. Instead, we opted to let the existing LSTM predict with either the action or trajectory representation by using the binary indicator to distinguish between these two operation modes. This choice allows us to share weights between the networks conducting the two tasks. Importantly, when doing visual servoing, we have to always set the value to indicate z, as it would otherwise plan with trajectories instead of actions. We added a comment further clarifying this to footnote 2.
>
> If you still feel that there are issues with the manuscript that would prevent you from raising your score, please point these out so that we can address them.

---

### Official Review · AnonReviewer3 · 2018-11-07
**Interesting paper with some clarifications required**

**Rating:** 7
**Confidence:** 4

**Review:**

The paper proposes a Variational IB based approach to learn action representations directly from video of actions being taken. The basic goal of the work is to disentangle the dynamic parts of the scene in the video from the static parts and only capture those dynamic parts in the representation. Further, a key property of these learned representations is that they contain compositional structure of actions so as to their cumulative effects. The outcome of such a method is better efficiency of the subsequent learning methods while requiring lesser amount of action label videos.

To achieve this, the authors start with a previously proposed video prediction model  that uses variational information bottleneck to learn minimal action representation. Next, this model is augmented with composability module where in latent samples across frames are composed into a single trajectory and is repeated in a iterative fashion and again the minimal representation for the composed action space is learned using IB based objective. The two objectives are learned in a joint fashion. Finally, they use a simple MLP based bijection to learn the correspondence between actions and their latent representations. Experiments are done on two datasets - reacher and BAIR - and evaluation is reported for action  conditioned video prediction and visual servoing.

- The paper is well written and provides adequate details to understand the flow of the material.
- The idea of learning disentangled representation is being adopted in many domains and hence this contribution is timely and very interesting to the community.
- The overall motivation of the paper to emulate how humans learn by looking at other's action is very well taken. Being able to learn from only videos is a nice property especially when the actual real world environment is not accessible.
- High Performance in terms of error and number of required action labeled videos demonstrates the effectiveness of the approach.

However, there are some concerns with the overall novelty and some technical details in the paper:
- It seems the key contribution of the paper is to add the L_comp part to the already available L_pred part in Denton and Fergus 2018. The trick use to compose the latent variables is not novel and considering that variational IB is also available, the paper lacks overall novelty. A better justification and exposition of novelty in this paper is required.
- Two simple MLP layers for bijection seems very adhoc. I am not able to see why such a simple bijection would be able to map the disentangled composed action representations to the actual actions. It seems it is working from the experiments but a better  analysis is required on how such a bijection is learned and if there are any specific properties of such bijection such that it will work only in some setting. Will the use of better network improve the learned bijection?
- While videos are available, Figures in the paper itself are highly unreadable. I understand the small figures in main paper but it should not be an issue to use full pages for the figure on appendix.
- Finally, it looks like one can learn the composed actions (Right + UP) representation while being not sensitive to static environment. If that is the case, does it work on the environment where except the dynamic part everything else is completely different? For example, it would be interesting to see if a model is trained where the only change in environment is a robot's hand moving in 4 direction while everything else remaining same. Now would this work, if the background scene is completely changed while keeping the same robot arm?

---

> ### Author Response · Authors · 2018-11-14
> **Clarified model choices, added experiment with background changes,**
>
> We thank the reviewer for the helpful comments and suggestions. We have updated the manuscript to address all points raised. In particular, we thank the reviewer for suggesting the experiment probing the model's invariance to background changes: we show that our model is robust to a variety of static backgrounds and believe this result strengthens the paper's argument. We address the reviewer's questions and comments as follows:
>
> 1. Q: Justification and exposition of novelty in the paper; prior work on composition?
> A: Thank you for your comment: we have revised the paper to justify and exposit the novelty of our work more clearly. We are not aware of prior work that uses composability as a loss, or that uses composition of this form in the context of video prediction or learning disentangled representations of action or dynamics. Please let us know if there is a relevant citation we have missed. The closest work we are aware of is Jaegle et al 2018, which was discussed in Section 2 in the original submission. Jaegle et al 2018 uses composition in conjunction with a siamese network to encourage motion representations. In contrast, our compositional loss encourages a latent space trained for prediction to be composable by a simple neural network; we show that this leads to disentangling of the action space. We have revised the manuscript to clarify our contribution: see the response to Reviewer 1 for details of the changes to the abstract and introduction.
>
> 2. Q:  Are the latent representations sensitive to background changes?
> A: As suggested by the reviewer, we ran an additional experiment to test whether our method indeed learns a representation disentangled from the background image. The results of this experiment are described in Section E of the Appendix. We train our method on reacher sequences with random background images: each sequence has a background image drawn from a subset of the CIFAR-10 training set. We test our network on sequences with backgrounds unseen during training (drawn from a held-out subset of the CIFAR-10 training set). Our network generalizes well to test sequences with unseen background images: it performs well on servoing (see Fig. 5, right and Fig. 9) and action sequences can be transplanted from one background image to another (Fig. 8). The fact that transplanted trajectories are completely synchronous with donor trajectories confirms that the learned latent properties are independent of the background.
>
> We performed an additional experiment that suggests that the action representation is also insensitive to changes in the agent’s appearance. Please refer to our response to Reviewer 2, and the corresponding experiment in Fig 8 for details. We believe these experiments provide strong evidence that the representation learned by our model is insensitive to static image content.
>
> 3. Q: Why choose a two-layer MLP for bijection? Why does this work?
> A: We chose a two-layer MLP because it is a very simple network: it has fewer parameters than deeper alternatives, making it less expressive, but easier to train effectively with a small amount of data. The goal of our method is to learn a representation that disentangles the structure of an agent's action space. If this goal has been met, a bijection between the latent space and the action space should be relatively simple to parameterize. Our method appears to learn such a representation, e.g., see the PCA plot in Fig. 6, included in the original submission. We find that a simple two-layer MLP converges quickly and achieves nearly perfect results. We did not find it necessary to use a more complex model to learn the bijective mapping for our full model.
>
> The compositional loss appears to be a key part of enabling a bijective mapping by making the action representation independent of static scene content. For models that do not capture the structure of the action space, such as the baseline without composability, the two-layer MLP failed to learn the bijection. We tried using a deeper and wider MLP in this case, but we were unable to produce any substantial improvements. We have clarified this point in Section B of the Appendix.
>
> 4. Q: Figures in the appendix are too small.
> A: For better visibility, we have truncated the length of the sequences displayed in Figures 7, 9, and 10 of the appendix (now numbered Figures 11, 13 and 14 in the revised manuscript) and enlarged the displayed images. The full sequences are still included on the website. We have also added a comment to the manuscript to encourage readers to view figures on a screen, where the sequences are easier to examine in detail.
>
> If you still feel that there are issues with the manuscript that would prevent you from raising your score, please point these out so that we can address them.

---

> > ### Comment · AnonReviewer3 · 2018-11-27
> > **Good Response and Revision**
> >
> > "We are not aware of prior work that uses composability as a loss, or that uses composition of this form in the context of video prediction..."
> >
> > - I agree with you, the comment was meant to convey that the overall concept of composing latent variables is not novel and has been applied in various fields. But it is indeed not used for video prediction and learning action dynamics tasks.
> >
> > - While individual components in the paper (IB, latent composition, etc.) are not novel in itself, the combination of them for the task of video prediction and learning action dynamics will be interesting to the community. Further, the revision done by the authors has certainly improved the quality of the paper and responses are satisfactory. Hence, I support this paper for acceptance.

---

### Author Response · Authors · 2018-12-10
**Authors' response summary: improved intro and abstract, added experiments on new environment variations, added technical clarifications**

We thank the reviewers for their helpful comments and suggestions: we have incorporated all proposed changes and performed all suggested experiments. We believe that the quality of the paper has been improved and our contribution is clearer. All reviewers agreed that the paper is well-motivated, well-written, and proposes an interesting and novel model. Reviewers 1 and 3 noted that the idea is relevant to the community and the experimental results are strong.

Generalization experiments. Reviewers 2 and 3 requested experiments probing the ability of the method to generalize to domains with different visual characteristics. To address this, we conducted experiments where the passively observed input domain contained the same degrees of freedom as the target domain but differed in terms of (i) the appearance of the background or (ii) the size and visual properties of the robot. Our experiments showed that the model trained on different source domains successfully generalizes to a target domain that has never been observed at training time. These results are shown in Figure 5 (right) and Figures 8 to 10 (in the appendix).  Reviewer 2’s main concern was whether the model can be applied when the source domain of passive learning and target domain for which action-labelled data is available are not the same: these experiments suggest that the model can be applied in such settings.

Clarifications. The reviewers further requested a number of clarifications to the paper. We have addressed all of the reviewers’ points in individual responses. We made the following changes to the manuscript in response to reviewer comments:
- Revised the abstract to more clearly emphasize the problem addressed in the paper and to make it clearer that our primary technical contribution is the introduction of the composition loss.
- Added a sentence to paragraph 1 of the introduction reiterating the problem we are addressing (learning an agent’s action space from unlabeled visual observations) and connecting it to the motivating example of an infant learning to walk.
- Added sentences to the third paragraph of the introduction connecting the title of the paper, the problem at hand, how our method specifically addresses this problem, and what our method can be used for.
- Added a clarification to Section 3.3, line 4: “from the training data” to clarify that the bijection is also trained on training data (not test data).
- To incorporate the generalization experiments requested by the reviewers (see above):
-- Added a final paragraph to Section 4.2 describing the experiments.
-- Expanded Table 2 to include these results, and moved it from the appendix into Figures 5 (right).
-- Added Figures 8, 9, and 10 (Appendix E) to display the results.
- Expanded table 1 to include data efficiency experiments suggested by reviewer 1.
- Added a sentence to footnote 2 to more clearly explain the choice to use a binary indicator to distinguish representations of single actions from representations of action sequences.
- Added a sentence to Appendix B mentioning that wider and deeper MLPs did not change performance in our experiments.
- Enlarged Figures 11, 13 and 14  (7, 9, and 10 in the original submission) by reducing the length of the displayed sequences. The full sequences are still available on the anonymized website associated with the paper.

---

### Meta-Review · Area_Chair1 · 2018-12-14
**boderline - but leaning to accept**

**Confidence:** 4
**Recommendation:** Accept (Poster)

**Metareview:**

The reviewers had some concerns regarding clarity and evaluation but in general liked various aspects of the paper. The authors did a good job of addressing the reviewers' concerns so acceptance is recommended.